



# Investigation of the October effect in VLF signals

Marc Hansen[1], Daniela Banyś[1], Mark Clilverd[2], David Wenzel[1] and M. Mainul Hoque[1]

[1]Institute for Solar-Terrestrial Physics, German Aerospace Center (DLR), Germany.
[2]British Antarctic Survey (UKRI-NERC), Cambridge, United Kingdom.

*Correspondence to*: Marc Hansen (Marc.Hansen@dlr.de)

**Abstract.** Subionospheric Very Low Frequency (VLF) radio signals are reflected by free electrons in the ionospheric D-region at about 60-90 km altitude and can propagate over long distances, which makes them useful for monitoring the state of the D-region or perturbations due to solar flares. At the D-region height, the ionosphere is mainly ionized by the solar Lyman-α radiation. The reflection characteristics of VLF signals depend on the state and dynamics of the D-region which is highly

influenced by the Lyman-α radiation. Although the amplitude of the received terrestrial VLF signal changes as a function of solar zenith angle over the course of the year, the VLF amplitude shows a distinctive sharp decrease around October, which is hence called the "October effect". This study investigates the occurrence of the October effect and its dependencies on latitude and longitude. We developed a method to detect the occurrence of the October effect in the long-term VLF data and derive key parameters characterizing (start and end date, intensity) the sudden decrease in the signal amplitude. This investigation

using a network of VLF stations distributed over low, middle and high latitude regions shows that the occurrence of the October effect has a clear latitudinal dependency, occurring earlier in high-latitude regions than at mid-latitudes. No low latitude signature is found.

## 1 Introduction

The ionosphere consists of several regions, of which the so-called D-region is the least ionized but the most complex in terms

of composition and a very dynamic layer. The D-region forms due to ionization by solar radiation (Nicolet and Aikin, 1960) at heights from about 60-90 km. The Lyman-α radiation photo-ionizes nitric oxide and galactic cosmic rays ionize all the other the neutral constituents. As a result, during the day the D-region is at a height of about 70 km and during the night at 85 km (Thomson, N. R. et al., 2017). This dependency on the Lyman-α radiation is also visible in the comparison between times of high and low solar activity, as a reduction in the solar activity leads to a decreased electron density and therefore a weaker

noon VLF amplitude (Thomson and Clilverd, 2000; Correia et al., 2011). In turn, an increase in solar activity leads to an increase in electron density (Thomson et al., 2021). However, at higher latitudes galactic cosmic rays become more significant for the ionization as the solar zenith angle (SZA) increases (Thomson. et al., 2017).
Subionospheric VLF signals propagate through the Earth-Ionosphere waveguide (Barr, 1971; Davies, 1990), with its upper boundary given by the D-region. The wave propagation is controlled by the index of refraction of the ionospheric medium and





thus controlled by the D-region electron density and the collision frequency of the electrons with the predominant neutral
       constituents (i.e., $N_2$ and $O_2$ (Hartree, 1931).

       The annual variation of the noon time D-region electron density shows a gradual increase in the first half of the year until it
       reaches a plateau in summer. In October a strong decrease is observed and the electron density reaches a low level in winter
(Renkwitz et al., 2023). These changes in the D-region electron number density also affect VLF propagation. During solar
       flare events (McRae and Thomson, 2004) and geomagnetic storms (Nwankwo, 2022) rapid temporal changes may occur in the
       D-region electron density and vertical structure (and thus collision frequency with neutral constituents) which influences VLF-
       propagation.

       Seen approximately from the middle of the year, the SZA at noon shows a symmetrical course over the year and therefore the
seasonal VLF signal amplitude behaviour at noon would also be expected to be symmetrical. Recently, Macotela, et al. (2021)
       found an asymmetry in the VLF signal amplitude noon curve, which is referred to as the fall effect. The VLF signal amplitude
       does not mirror the variation of the SZA and tends to remain at mid-summer levels or increase somewhat before experiencing
       a strong and sudden decrease around October. This decrease in October has also been reported by Banyś (2017). We investigate
       the strong and sudden decrease at the beginning of October, which we refer to as the October effect. The term "October effect"
was first used by Pancheva and Mukhtarov (1996) for the sharp transition of the electron density profile in the lower D-region
       in autumn, which in turn affects VLF propagation. In addition to that, recently Wendt et al. (2023) found that there is no
       October effect at night.

       The October effect is shown in Fig. 1 for three propagation paths at different latitudes together with the solar elevation angle
       (SEA) observed at the mid reflection point (MRP). The SEA is the opposite angle of the SZA. Figure 1 a) shows the location
of the propagation paths and their corresponding transmitter (Tx), receiver (Rx) and MRP (black dot). Figure 1 b) shows the
       VLF composite signal amplitude at noon for the propagation path NAA – SOD (MRP at 64°N) with an early and strong October
       effect (marked by vertical broken lines) and Fig. 1 c) shows NAA – ESK (MRP at 54°N) with a later October effect.
       Additionally, Fig 1 b) shows a decrease in VLF amplitude prior to the October effect. Finally, Fig. 1 d) shows the VLF
       amplitude for the propagation path from NAU – STJ (MRP at 33°N) and it shows no October effect. The VLF amplitude
follows the overall trend of the SEA.

       The investigation of the October effect´s dependencies will help to better understand the course of the VLF signal amplitude
       fluctuations. As pointed out in Banyś (2017), considering the background VLF signal amplitude is crucial to correlate the VLF
       signal response to the strength of a solar flare. In order to identify the physical origin of the October effect, we attempt to
       quantify its characteristics and determine its variation with geographic latitude and longitude. The paper is structured as
follows: First, the data sources and used propagation paths are presented in section 2. Then, the method for determining the
       parameters of the October effect is introduced in section 3. With that, we obtain clear results in section 4 for determining the
       dependency of the October effect on latitude on longitude. The results are then discussed in section 5, and our conclusions are
       summarized in section 6.





## 2 Data

The data used in this study are made available by the Antarctic-Arctic Radiation-Belt (Dynamic) Deposition - VLF
Atmospheric Research Konsortia (AARDDVARK) (Clilverd et al., 2009) network of the British Antarctic Survey (BAS) and
the DLR´s Global Ionospheric Flare Detection System (GIFDS) (Wenzel et al, 2016) network. The available datasets include
either 1 Hz or 10 Hz relative measurements of the signal amplitude and phase, depending on the receivable transmitters and
used receiver types. The receiver stations of the AARDVARK network are mainly located at high latitudes while GIFDS
receiver stations are located at mid latitudes. For a detailed description of the technical background of GIFDS we refer D.
Banyś (2017). This work focusses on the analysis of the amplitude data. The used transmitter (Tx) – receiver (Rx) combinations
and their corresponding frequencies, locations of Tx/ Rx and their MRP are shown in Table 1.

**Figure 1:** Map of used great circle propagations paths in a). Tx (triangle up), MRP (dot), Rx (triangle down). $A_{day}$ (black) ± σ (grey) with the SEA (red) for 3 exemplary VLF links: b) NAA – SOD, c) NAA – ESK and d) NAU – STJ. $A_{day}$ are composites of different years, due to different data availability: for NAU – STJ 2014, 2016 and 2019, for NAA – ESK 2016-2019 and for NAA – SOD 2013, 2015-2016 and 2021 are used. The relevant time period for the October effect in light grey. The determined $t_{start}$ and $t_{end}$ of $A_{day}$ as black vertical dashed lines in b) and c). $A_{day}$ in d) shows no October effect.





The used propagation paths are illustrated in their corresponding maps. The following VLF receiver stations data are used in the analysis and their abbreviation are given in the parenthesis: Eskdalemuir (ESK), Neustrelitz (NTZ), St. Johns (STJ), Kilpisjärvi (KIL), Sodankylä (SOD) and Ny-Ålesund (NYÅ). The transmitters are represented by their call signs. The used propagation paths are mainly over North America and the Atlantic. The signals originate mainly from US American transmitter stations and are received in Europe. The used Tx-Rx combinations are chosen due to their comparability in latitude and longitude and availability of continuous measurements over a long time period. Furthermore, only years of propagation paths without huge data gaps or jumps in the amplitude signal level are considered, as these jumps could indicate a change of receiver hardware.

**Table 1:** Overview of all used transmitter-receiver combinations.

| Tx-Rx | Tx lat, long | Rx lat, long | MRP geographic lat, long [°] | MRP geomagnetic lat, long [°] | $f$ [kHz] | $d$ [km] |
|---|---|---|---|---|---|---|
| **Used in Introduction (ordered in decreasing geographic latitude):** | | | | | | |
| NAA-SOD | 44.64, -67.28 | 67.42, 26.59 | 64.22, -38.08 | 66.82, 50.55 | 24.00 | 5664.33 |
| NAA-ESK | 44.64, -67.28 | 55.27, 3.18 | 54.47, -39.19 | 56.48, 46.19 | 24.00 | 4562.09 |
| NAU-STJ | 18.40, -67,18 | 47.57, -52.71 | 33.19, -61.17 | 38.27, 20.21 | 44.75 | 3496.51 |
| **Investigation of latitudinal dependency (ordered in increasing geographic latitude):** | | | | | | |
| NAA-ESK | 44.64, -67.28 | 55.27, 3.18 | 54.47, -39.19 | 56.48, 46.19 | 24.00 | 4562.09 |
| NAA-NTZ | 44.64, -67.28 | 53.35, 13.07 | 56.33, -31.32 | 57.06, 53.79 | 24.00 | 5624.38 |
| NRK-STJ | 63.85, -22.47 | 47.57, -52.71 | 56.60, -40.83 | 58.86, 44.74 | 37.50 | 2579.95 |
| NAA-KIL | 44.64, -67.28 | 69.02, 20.89 | 63.76, -40.95 | 67.35, 48.45 | 24.00 | 5380.62 |
| NAA-SOD | 44.64, -67.28 | 67.42, 26.59 | 64.22, -38.08 | 66.82, 50.55 | 24.00 | 5664.33 |
| NDK-KIL | 46.37, -98.34 | 69.02, 20.89 | 70.02, -67.10 | 77.00, 19.69 | 25.20 | 6265.59 |
| NDK-SOD | 46.37, -98.34 | 67.42, 26.59 | 71.04, -64.53 | 77.34, 23.53 | 25.20 | 6557.68 |
| NRK-NYÅ | 63.85, -22.47 | 78.92, 11.93 | 72.04, -12.20 | 71.66, 79.59 | 37.50 | 2007.25 |
| **Investigation of longitudinal dependency (ordered in increasing geographic longitude):** | | | | | | |
| NLK-NYÅ | 48.20, -121.92 | 78.92, 11.93 | 72.30, -107.35 | 78.75, -51.33 | 24.80 | 5560.93 |
| NLK-KIL | 48.20, -121.92 | 69.02, 20.89 | 75.37, -92.33 | 83.47, -30.58 | 24.80 | 6629.52 |
| NLK-SOD | 48.20, -121.92 | 67.42, 26.59 | 76.72, -91.30 | 84.62, -29.61 | 24.80 | 6892.37 |
| NDK-NYÅ | 46.37, -98.34 | 78.92, 11.93 | 69.16, -82.20 | 77.32, -5.83 | 25.20 | 5386.70 |
| NDK-KIL | 46.37, -98.34 | 69.02, 20.89 | 70.02, -67.10 | 77.00, 19.69 | 25.20 | 6265.59 |
| NDK-SOD | 46.37, -98.34 | 67.42, 26.59 | 71.04, -64.53 | 77.34, 23.53 | 25.20 | 6557.68 |
| NAA-NYÅ | 44.64, -67.28 | 78.92, 11.93 | 65.40, -53.10 | 70.46, 35.28 | 24.00 | 4931.98 |
| NAA-KIL | 44.64, -67.28 | 69.02, 20.89 | 63.76, -40.95 | 67.35, 48.45 | 24.00 | 5380.62 |
| NAA-SOD | 44.64, -67.28 | 67.42, 26.59 | 64.22, -38.08 | 66.82, 50.55 | 24.00 | 5664.33 |
| NRK-NYÅ | 63.85, -22.47 | 78.92, 11.93 | 72.04, -12.20 | 71.66, 79.59 | 37.50 | 2007.25 |



## 3 Method

Data processing is crucial to derive meaningful conclusions. Since the measurements are relative and each receiver station is unique it is challenging to formulate a general approach to the data and to make comparisons between the propagation paths.

Therefore, we propose a detection method based on the derivatives and therefore the data doesn't require further preprocessing. If comparable composites of data are required, a recent work by Schneider et al. (2023) provides a clear overview. In Fig. 2, we have considered the propagation path of NAA-NYÅ in 2021 and we describe the process of obtaining the key parameters of the October effect: the start date $t_{start}$, the end date $t_{end}$, the date of maximum amplitude decrease $t_{max}$ and intensity $m_{Oct}$ of the October effect. First, we computed the median of VLF amplitude $A$ for every 10 min interval for one entire year

separately for each propagation path from the raw VLF amplitude measurements (see Fig. 2 a). In the next step we define the local noon time as the time of the maximum SZA at the MRP of the propagation path. At this local noon time we set a time windows of $\pm 1$ h to derive the VLF amplitude noon curve $A_{day}(t)$ from $A(t)$ as a rolling median of 21 days.

From $A_{day}(t)$ the first derivative $dA_{day}/dt$ and the second derivative $dA_{day}^2/dt^2$ are calculated (see Fig. 2 c) and d)). We use the minimum and maximum in $dA_{day}^2/dt^2$, to define the start $t_{start}$ and the end $t_{end}$ of the October effect (see the red points

in Fig. 2 b)). Additionally, the time of the maximum decrease of the October effect $t_{max}$ is found by getting the zero crossing $dA_{day}^2/dt^2 = 0$. $t_{max}$ also marks the point of the lowest gradient in $A_{day}$ and is the local minimum in $dA_{day}/dt$. From $t_{start}$ and $t_{end}$ the duration of the October effect is easily computed by $\Delta t = t_{end} - t_{start}$. To obtain a quantity of the intensity of the October effect the slope of $A_{day}$ is used. The most northern propagation paths may not be completely in daylight during the winter, which leads to an uncertainty in finding $t_{end}$. Therefore, the slope is calculated from $t_{start}$ to $t_{max}$, thus the

intensity of the October effect is derived by $m_{Oct} = (A_{max} - A_{start})/(t_{max} - t_{start})$ .

Missing data and data within noise level may lead to false interpretation, for example the regular vertical dark blue stripes in the day time are from maintenance downtimes of the NAA transmitter. Therefore, $dA_{day}/dt$ is smoothed by a gaussian filter. The method we use to calculate $A$ and $A_{day}$ considers NaNs in the data as missing data and the median then gets calculated from fewer data points or is considered NaN when there are no data points. Adding to that, we formulate certain criteria for

the determination of the October effect:

1.  $t_{start}$ and $t_{end}$ need to lie in the same minimum in $dA_{day}/dt$, see Fig. 2 c). If this is not the case, the algorithm searches for the next set of $t_{start}$ and $t_{end}$ by limiting the time window of the search.

2.  No huge spikes $dA_{day}/dt > 2.5$ dB/day to avoid receiver stations where a hardware change has occurred. This would be visible due to a change in the overall level of $A_{noon}$.

3.  $dA_{day}^2/dt^2$ at $t_{start}$ and $t_{end}$ needs to be bigger than 0.005 $dB/day^2$, as seen as a dotted horizontal line in Fig. 2 d). This is to ensure that a significant decrease happens around October, thus the October effect occurs.

4.  $\Delta t$ must be longer than 7 days to consider outages in the transmitter, which could be identified as sharp decreases.

5.  Between $t_{start}$ and $t_{end}$ there need to be more days with data points than $\Delta t/2$.



6.  The algorithm searches in a time window from the 5. September to 15. November. $t_{start}$ and $t_{end}$ should differ
    from these limits.



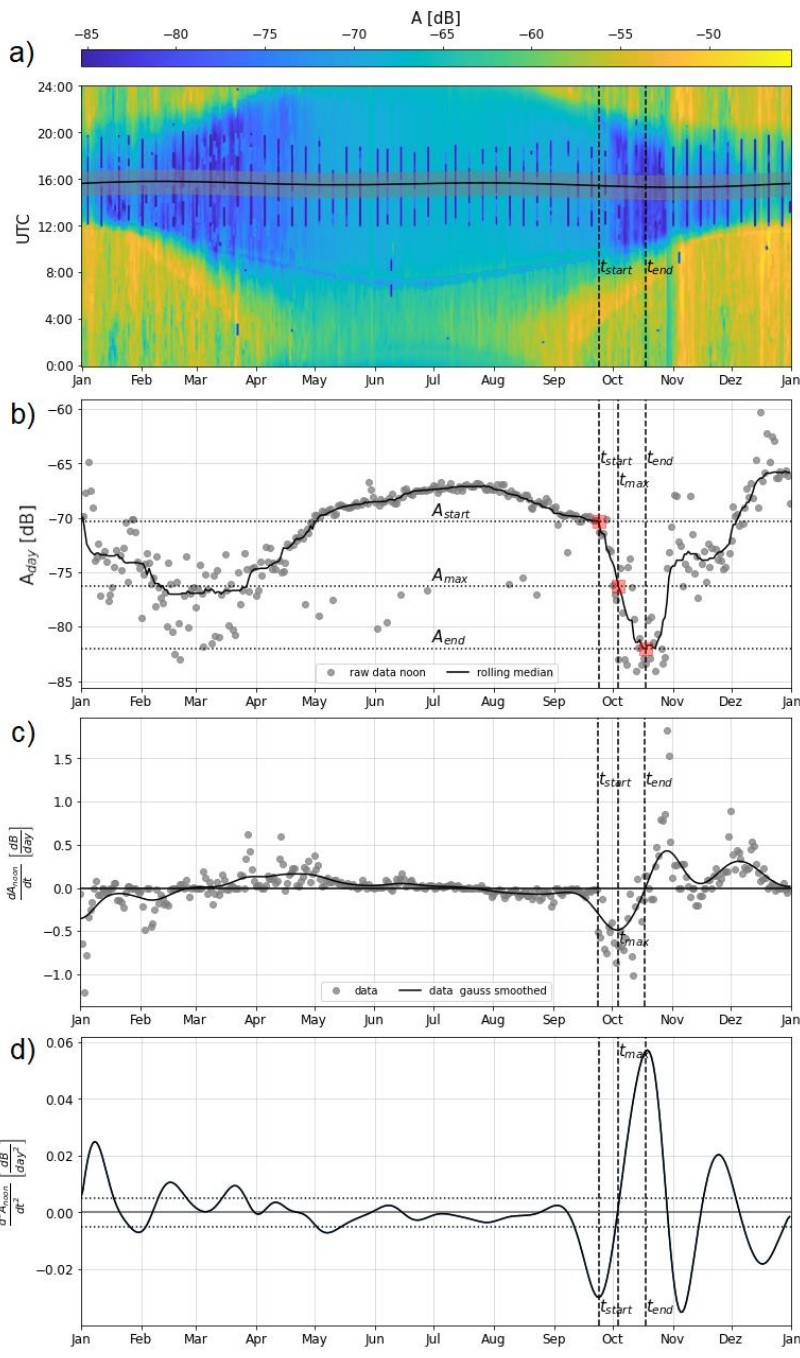

**Figure 2.** Steps for deriving $t_{start}$, $t_{max}$ and $t_{end}$ of the October effect: a) Smoothed VLF amplitude over the year for each day with local noon time window (shaded). b) deduced noon curve $A_{day}$ (solid) with $t_{start}$, $t_{max}$ and $t_{end}$ (dashed) and associated values $A_{start}$, $A_{max}$ and $A_{end}$ (dotted). c) first derivative $dA_{day}/dt$, and d) second derivative $\mathrm{d}A_{day}^2/\mathrm{dt}^2$ defining $t_{start}$, $t_{max}$ and $t_{end}$ if threshold (dotted) is exceeded.





## 4 Results

Next, we investigate how the October effect varies from year to year and with longitude and latitude. For this purpose, a careful selection of Tx/Rx-pairs is necessary. For studying the variation of the October effect with longitude, propagation paths with
the same latitude related to their MRP are selected. Likewise, propagation paths with the same longitude related to their MRP are selected for investigating the latitudinal dependency.

### 4.1 Year-to-year variability

First, the year-to-year variation is investigated and the results are show in Fig. 3. To distinguish the different propagation paths,
they are color coded according to the latitude or longitude of the mid reflection point. This is shown in Fig. 3 a) and e). The variation of $t_{start}$ is shown in Fig 4 b) and f), $t_{end}$ is shown in Fig. 3 c) and g) and the intensity $m_{Oct}$ is shown in Fig. 3 d) and h). Comparing the different propagation path over time, a clear trend in $t_{start}$, $t_{end}$ or $m_{Oct}$ of the October effect is not visible, but there is a significant variation over the years. This points to a highly variable D-region which is driven by both variations in ionization and dynamics of the neutral atmosphere. From the color coded $t_{start}$ in Fig. 3 b) and f) and $t_{end}$ in Fig.
3 c) and g) of the October effect, a possible latitudinal and longitudinal dependency is visible, as the October effect occurs earlier in higher latitudes and more easterly longitudes – over the limited longitudinal range studied.. This is investigated further in the next subsection.

### 4.2 Latitudinal dependency

To investigate the latitudinal dependence of the October effect, 8 propagation paths NAA-SOD/KIL/ESK/NTZ and NRK-STJ/
NYÅ and NDK-SOD/KIL are used, as their MRPs are around 40°W. From $t_{start}$ in Fig. 4 b) it is evident that the October effect occurs earlier at higher latitudes. Likewise, from the variation of $t_{end}$ in Fig. 4 b) we see that the October effect also ends earlier with increasing latitude. Interestingly, the duration $\Delta t$ stays constant. The intensity $m_{Oct}$ in Fig. 4 c) shows a wider spread at higher latitudes, which could point to less solar forcing of the October effect and therefore its dependency on the dynamics of the atmosphere.

It is important to point out that in Fig. 4 a) to c) the two additional propagation paths NDK-SOD/KIL are shown. For these paths the October effect occurs even earlier in the year, which suggests that not only the higher latitude but also being in the American sector could be relevant. This is further supported by the additionally shown propagation path NRK-NYÅ. The MRP for this propagation path has about the same latitude ~70°N as NDK-SOD/KIL, but the October effect occurs later. This could be contributed to the more eastern longitude.

Considering the geomagnetic latitude, which is shown in Fig. 5 b), a linear trend in the latitudinal dependency is visible. Also, $t_{start}$ and $t_{end}$ of NRK-NYÅ now align with the overall trend. The geomagnetic latitude of its MRP is about 71.6°N, while NDK-SOD/KIL are about 77.6°N in geomagnetic latitude.



**Figure 3.** Yearly variations of the October effect: a) and e) show maps with the used propagation paths. On the left (a-d) is the yearly variation of the latitudinal dependency investigation colour coded in latitude and the right (e-h) the propagation paths for the longitudinal dependency investigation colour coded in longitude. b) and f) show $t_{start}$, c) and g) $t_{end}$, d) and h) the intensity $m_{Oct}$ of the October effect.



### 4.3 Longitudinal dependency

To investigate the longitudinal dependency the 10 propagation paths NLK-SOD/KIL/NYÅ, NDK-SOD/KIL/NYÅ, NAA-SOD/KIL/NYÅ and NRK-NYÅ are selected. While their MRPs are all at comparable latitudes of about 65°N to 75°N, they differ greatly in longitude from 110°W to 10°W. Fig. 4 d) shows the great circle paths and their corresponding MRPs, marked as black dots. It should also be noted, that although these propagation paths have different ranges, we considered them all as long propagation paths.

In Fig. 4 e) and f) the results for $t_{start}$, $t_{end}$ and $m_{Oct}$ are shown. In the American sector the October effect occurs earlier and also ends earlier. Again, the duration $\Delta t$ stays about the same. Also, the intensity $m_{Oct}$ does not vary with longitude, rather, it is scattered over a wide range of values. This is in contrast to the latitudinal dependency of $m_{Oct}$.

It is important to point out, that the latitude, which also has an effect on $t_{start}$ and $t_{end}$, is slightly different for these propagation paths, but these are the best paths to compare in longitude, as the possibilities are very limited. Adding to this

point, it should be considered that these propagation paths span a wide range of longitudes and therefore the longitudinal dependency might be smeared out over these long paths. Unfortunately, there are no paths that run meridional at different longitudes, as these would possibly be best for such a comparison. This is in contrast to the investigation of the latitudinal dependency, as the used propagation paths there have a more zonal propagation direction.

Another point is, that the most eastern propagation path from NRK-NYÅ differs from the general trend of $t_{start}$ and $t_{end}$.

One reason for this might be, that the propagation path is much shorter and that the transmitter NRK has a higher frequency of 37.5 kHz, thus being in the Low Frequency (LF), because the VLF frequency range is defined as the frequency range of 3-30 kHz. But as NRK-NYÅ aligns well in linear trend the investigation of the geomagnetic latitudinal dependency (see Fig. 5 d)), the higher frequency and the shorter propagation path do not seem to be a main influence here.

It should be noted that the most westward propagation path NLK-NYÅ also does not follow the general trend and the

October effect occurs later in the year. This "s-shaped" behaviour of the longitudinal dependency of the October effect suggests a connection to the distance from the auroral oval, as the shape and extension of the auroral oval differs at different longitudes.

Here the direct comparison to the geomagnetic longitude shows no significant change in the overall trend and the "s-shaped" behaviour is still visible, as it can be seen in Fig. 5 d).

## 5 Discussion

VLF signal amplitude measurements conducted by the AARDDVARK and GIFDS networks from the years 2006 to 2022 for different propagation paths (see Tab. 1) show a distinct sudden sharp decrease in the VLF noon amplitude $A_{day}$ (see Fig. 1) around October. This sudden decrease is called the October effect and does not have a satisfactory explanation yet. The variation of $A_{day}$ in Fig. 1 would be expected to follow the overall symmetrical course of the SZA over the year, as the F10.7

irradiance from the Sun is the main driver of the electron density in the ionospheric D-region.





**Figure 4.** Latitudinal and longitudinal dependency of the October effect: a) and d) show maps with the used propagation paths. On the left (a-c) is the latitudinal dependency colour coded in longitude and on the right (d- f) the longitudinal dependency colour coded in latitude. b) and e) show the $t_{start}$ (triangle up) and $t_{end}$ (triangle down) of the October effect. The triangles are at the median and the lines show the spread of the values. c) and f) show the intensity $m_{Oct}$ of the October effect.



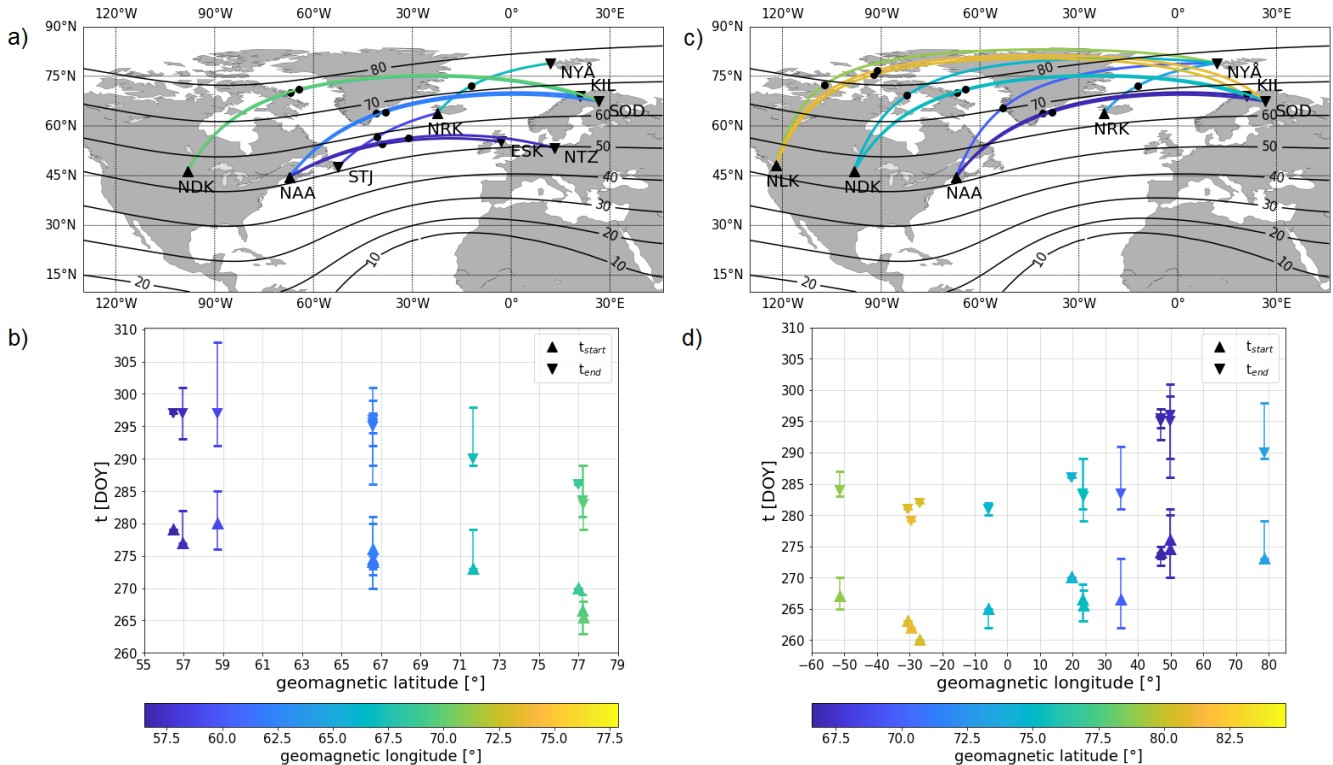

**Figure 5.** Dependency of the October effect on geomagnetic latitude and longitude: a) and c) show maps with the used propagation paths. On the left (a-b) is the geomagnetic latitudinal dependency colour coded in geomagnetic longitude and the right (c- d) the geomagnetic longitudinal dependency colour coded in geomagnetic latitude. b) and d) show the $t_{start}$ (triangle up) and $t_{end}$ (triangle down) of the October effect. The triangles are at the median and the lines show the spread of the values.

In this study we presented a method (section 3, Fig. 2) to detect the October effect in the seasonal variation of the VLF amplitude $A_{day}$. With this method we quantified the start $t_{start}$, the end $t_{end}$, its intensity $m_{Oct}$ and the duration $\Delta t$ of the October effect. A strong yearly variation in $t_{start}$ and $t_{end}$ is visible in Fig. 3 b), c), f) and g). This yearly variation shows a latitudinal and longitudinal dependency of $t_{start}$ and $t_{end}$. The results presented here suggest two cases for further investigation: one with propagation paths, where the MRP is at a similar longitude range to isolate the latitudinal dependency and the other case where the MRP is at a similar latitudinal range to investigate the longitudinal dependency.

A challenge in detecting the October effect is the determination of $t_{end}$, as the VLF amplitude in winter begins to rise again and also shows a stronger variation in some propagation paths. Therefore, the end point is not always clearly determined, especially in high latitudes where parts of the propagation paths don't lie in daylight during winter. This also has an effect on



the determination of the duration $\Delta t$ of the October effect. For the future we also plan to investigate the October effect in the southern hemisphere, although even less data is available for this.

We also investigated the dependency of the October effect on the solar activity, but we did not identify conclusive results. As we hypothesize that the October effect is connected to the dynamics of the neutral atmosphere, more solar forcing of the D-region should result in a weaker October effect. As an example of this solar forcing Clilverd et al. (2006) showed the ionospheric effective height is lowered during solar proton events (SPEs), which occurs when the Sun is active. The weaker October effect is sometimes visible in $A_{day}$, where the sudden decrease in $A_{day}$ is more gradual than usual. Also, a wider spread of $m_{Oct}$ was observed during times with lower solar activity for some propagation paths than in times with high solar activity. Additionally, that Wendt et al. (2023) found an increase in neutral atmospheric temperature during the October effect, which supports the connection between the October effect and the dynamics of the neutral atmosphere. This will be a topic for our ongoing work.

## 6 Conclusion

From the results in Fig. 4 and Fig. 5, it can be concluded that the October effect is dependent on the latitude and also shows systematic variations over a limited range of longitude. It occurs earlier in higher latitudes and the spread of the intensity of the October effect ($m_{Oct}$) is also larger in that region. Additionally, it occurs earlier in the American sector than the European sector. The longitudinal dependency of $t_{start}$ and $t_{end}$ of the October effect shows a distinct "s-shaped" behavior. This "s-shaped" behavior points strongly to an association with the distance from the auroral oval. When $t_{start}$ and $t_{end}$ are considered in the terms of the magnetic latitude and longitude, a clear linear dependency on geomagnetic latitude is visible, while the s-shaped behaviour of the longitudinal dependency continues to be visible in the geomagnetic longitude. Adding to that, as spread of the intensity $m_{Oct}$ is larger in higher latitudes, where less solar forcing occurs, the neutral atmosphere and its dynamic nature appears to be the main driver of the October effect.

**Data availability** The used AARKVARK VLF data can be accessed at:https://psddb.nerc-bas.ac.uk/data/access/coverage.php?menu=4,7&bc=1&source=1&class=284,37,140,255,243,3,110,232,141,30,279&type=ULTRA. GIFDS VLF data can be provided by the corresponding authors upon request.

**Author contribution**: MH performed the systematic analysis of the VLF measurements, the investigation of the dependencies and the visualization of the results. DB, MC and DW provided supervision. MH and DB wrote the manuscript draft. MC, MMH and DB reviewed and edited the manuscript.

**Competing interests**: The authors declare that they have no conflict of interest.

**Acknowledgements:** This work is supported by "AMELIE - Analysis of the MEsosphere and Lower Ionosphere fall Effect" (DLR project D/921/67286532).

We thank all members of the AARDDVARK network, the UK Polar Data Centre (PDC), Natural Environment Research





Council (NERC) and the British Antarctic Survey (BAS) for providing the data throughout and for their willingness to share

it publicly.

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
