# Peer review of "Investigation of the October effect in VLF signals"

_Annales Geophysicae, 2023_

## Referee Comment (RC1)

Revieuw of: **Investigation of the October effect in VLF signals**, by Marc Hansen et al submitted to *Annales geophisicae*.

The paper study focuses on the October effect using the VLF signal propagation technique. In its form, the paper presented materials are not convincing and should be revised. So I reject the publication of the paper.

-) In the data processing, it is not clear what the authors did when the noon time coincided with ionospheric disturbance times such as solar flares, lightning, early events, geomagnetic storms…

-) The color code used in the presented results are not clear and from figure to other the color code changes. For example, in figure 3 the NDK-KILL path is cyan colored but in figure 4 is violet. I recommend unifying the color coding for each path and using more forms not only two filled triangles. Also, use open forms rather than filled forms.

---

## Author Response (AR1)

**Anonymous Referee #1:**

*Comment 1: "In the data processing, it is not clear what the authors did when the noon time coincided with ionospheric disturbance times such as solar flares, lightning, early events, geomagnetic storms…"*

Answer 1: To address the concerns about what happens when the noon time coincides with ionospheric disturbance times, we added a paragraph explaining the connection and clarify that these events are negligible due to the calculation of the median noon amplitude per day and the overall smoothing with a longer time period of 21 days. (Lines 93-103)

*Comment 2: "The color code used in the presented results are not clear and from figure to other the color code changes. For example, in figure 3 the NDK-KILL path is cyan colored but in figure 4 is violet. I recommend unifying the color coding for each path and using more forms not only two filled triangles. Also, use open forms rather than filled forms."*

Answer 2: Thank you for noticing this ambiguity. The color code used in the presented results is confusing, we changed the color coding in a way to unify the illustrations, while still showing the different dependencies of latitude and longitude. To make the different coloring clear and more visible, we choose to used different colormaps for the latitude and longitude dependency. As some latitudes or longitudes are close to each other, the corresponding propagation paths also have similar colors. While this could lead to problems in distinguishing different propagation paths, we still choose to use it to ensure the conformity with the colorbar and its monotonically increase in color. Furthermore, we switched to open forms to further improve clarity.

**Anonymous Referee #2**

*Comment 1: "In order to clarify some needed details about the data and methods (addition X, gamma influence on VLF, etc.) I suggest the authors to create a github project with them."*

Answer 1: To further clarify the influence of x-ray and gamma radiation on the propagation of VLF waves, we added a paragraph explaining the connection and clarify that these short-term events are negligible due to the calculation of the median noon amplitude per day and the overall smoothing with a longer time period of 21 days. (Lines 93-103)

The used AARDDVARK VLF data is publicly available and can be accessed at: https://psddb.nerc-bas.ac.uk/data/access/coverage.php?menu=4,7&bc=1&source=1&class=284,37,140,255,243,3,110,232,141,30,279&type=ULTRA.

As data from the GIFDS network is not publicly available, we uploaded the used GIFDS data as a Supplement to the paper.

As the method itself relies on functions from the Pandas or NumPy Python libraries, we have refrained from explaining the calculation further.

*Comment 2: "In Sec. 6. Conclusion I suggest the authors to present the most important conclusions as a bulleted list or similar."*

Answer 2: Thanks for your advice to better get our conclusions across. We added a bulleted list to the end of the Conclusion section to highlights our three most important findings.
* * *
Please be also aware that we have added Tero Raita as a co-author, as he manages the VLF Receiver stations in SOD and KIL, which we use in our data analysis.

Additionally, we have added some references for the revision of the method section and, moreover, some recent publications.

With kind regards,

The authors

---

## Author Response (AR2)

**Report 1**

The authors properly answered all reviewers' questions. There are only minor revisions to be done before being acceptable for publication.

1. Line 97: change 'evets' by 'events'

2. Line 164: 'marked as black dots' I think must be 'marked with dots in the same line color'

3. Figure 4: the labels 'e' and 'd' must be changed. (d) is the map

We thank the reviewer for the report and spotting of these errors. We have addressed all the corrections in the text.

**Report 2**

**Comment 1**

In the methodology, the authors were not criterios on the possible effects of the magnetic storm in the ionosphere and, consequently, in the propagation of the VLF wave in this medium under these particular conditions. For example, they stated "During geomagnetic 100 storms the VLF amplitude may be decreased for a couple of days...", however, during a geomagnetic storm two different events occur in the ionosphere: dirtuebed dynamo and prompt penetration electric field. Although those effects are predominant in the F region. but due to the connection between the E and F region though the magnetic field lines, it can reflect the lower part of the ionosphere and the consequences must be different. I am also concerned about how the solar x-ray burst could interfere in the results if they occur around noon? I have a suggestion, how could be the result affected if the authors removed the disturbed period from the data? Maybe, removing those days, the results must be more convincing.

**Answer 1**

To investigate to influence of geomagnetic storm on the VLF propagation, we took the Kp indices from GFZ Potsdam https://www.gfz-potsdam.de/en/section/geomagnetism/data-products-services/geomagnetic-kp-index, which are given for each 3 h interval of a day and the NOAA definition of Kp smaller than 5 as undisturbed (https://www.swpc.noaa.gov/noaa-scales-explanation). In the next step we took the Kp index for that day and time interval for each link at the local noon of the mid reflection point (MRP) of the propagation path. Therefore, we derive the following plots, which show the course of the Kp index over the DOY (Day of Year) during the occurrence of the October effect. As the October effect occurs during local noon, which is also the time we took the VLF amplitude data, we ensure with that, that we compared each propagation path to the right corresponding Kp index.

From these plots, it becomes clear that only 5 days in the whole data set are disturbed during the October effect and only for 1 day. We did this analysis for all used paths in the dataset, but we show here only the paths, where the Kp index is greater 5. The dates with the affected propagation paths are:

As it was pointed out in Nwanko, 2022, during a geomagnetic storm the VLF may be disturbed, but it is not clear if the VLF amplitude is affected or not. Adding to that, all the Kp indices for these days are in the range of G1 (minor)/G2 (moderate) NOAA classification. To investigate the impact in the next step, we compared the VLF measurement for the disturbed day and one day before and after, to see, if the amplitude level at local noon of the MRP shows a distinct reduction. Here, for each of the propagation paths with a potential disturbance the course of the Kp index during the October effect is shown together with the potential disturbed daily amplitude. With that, we investigate each pair and discuss possible implications:

| Tx | Rx | Year | DOY | Kp |
|-----|-----|------|-----|-------|
| NLK | NYA | 2008 | 285 | 6.333 |
| NAA | SOD | 2012 | 287 | 5.667 |
| NDK | SOD | 2015 | 280 | 6.333 |
| NDK | NYA | 2015 | 280 | 6.333 |
| NDK | SOD | 2016 | 271 | 5.667 |
| NDK | KIL | 2016 | 271 | 5.667 |
| NDK | NYA | 2016 | 271 | 5.667 |
| NRK | STJ | 2017 | 286 | 5.667 |
| NAA | SOD | 2017 | 286 | 5.667 |
| NAA | NTZ | 2017 | 286 | 5.667 |

Table 1: Propagation paths with the corresponding time and Kp value of the disturbed time.

**2008**

From the course of the Kp index in figure 1 left, it becomes clear that Kp is greater 5 on the 285th day. Although, the VLF daily amplitude for that day does not show a significant disturbance. Therefore, the determination of the October effect should not be influenced by it.

[Figure]

Figure 1: Left: The course of the Kp index during the duration of the observed October effect in that propagation path. Right: The daily amplitude one day prior (blue), at the day of the enhanced Kp index (black) and one day after (orange).

**2012**

From the course of the Kp index in figure 2 top/bottom left, it becomes clear that Kp is greater 5 on the 287th day. The VLF daily amplitude (right) shows a disturbance with a lowered VLF signal amplitude on that day for both propagation paths. As this disturbance is in the middle of October effect, the determination of the ending and starting of the October effect, is not affected.

[Figure]

Figure 2: Left: The course of the Kp index during the duration of the observed October effect in that propagation path. Right: The daily amplitude one day prior (blue), at the day of the enhanced Kp index (black) and one day after (orange).

**2015**

From the course of the Kp index in figure 3 left, it becomes clear that Kp is greater 5 on the 280th day of the October effect. For NDK-NYA this falls at the last day of the October effect, which could affect the determination of the ending time. But the VLF amplitude on that day, shown in figure 3 top right, does not show a significant change and therefore the determination of the October effect should not be affected. For NDK-SOD in figure 3 middle and bottom right it is hard to say how the disturbance is affecting the VLF signal, as a solar flare at around 14:00 can be seen. Although, the VLF overall signal level does not seem disturbed, if compared the signal level prior to the flare and after. Adding to that, the disturbance does not occur at the end of the October effect and therefore the determination of it, should not be affected.

[Figure]

Figure 3: Left: The course of the Kp index during the duration of the observed October effect in that propagation path. Right: The daily amplitude one day prior (blue), at the day of the enhanced Kp index (black) and one day after (orange).

**2016**

From the course of the Kp index in figure 5 left, it becomes clear that Kp is greater 5 on the 271th day. Also a clear reduction in all the VLF signal on that day is visible on the right side of figure 5. But the disturbance is on the 271th day and therefore later than the beginning of the October effect. From that, a change in the determination of the beginning and end of the October effect is highly unlikely. Only the propagation path of NDK-KIL is a special case here, as the October effect is determined to begin at the 270th, the disturbance on the 271th is therefore in close proximity. From the paper figure 4 shows the dependency of the October effect on latitude and longitude. The propagation path of NDK-KIL in 2016 is there visible as one data point on latitude 70° and $t_{start}$ of 270. This point is just a little bit later than the general trend. As for the dependency on longitude, the same data point is visible at again DOY 270 and longitude of about -68 °. Here, it again is a little later than the general trend might expect. This could be contributed to the disturbance. But as this is only one point in the whole dataset, the overall statement concerning the timing of the October effect in dependency of latitude and longitude is not affected.

[Figure]

Figure 4: Latitudinal and longitudinal dependency of the October effect: a) and d) show maps with the used propagation paths. On the left (a-c) is the latitudinal dependency colour coded in longitude and on the right (d- f) the longitudinal dependency colour coded in latitude. b) and e) show the $t_{start}$ (triangle up) and $t_{end}$ (triangle down) of the October effect. The triangles are at the median and the lines show the spread of the values. c) and f) show the intensity $m_{Oct}$ of the October effect.

[Figure]

Figure 5: Left: The course of the Kp index during the duration of the observed October effect in that propagation path. Right: The daily amplitude one day prior (blue), at the day of the enhanced Kp index (black) and one day after (orange).

**2017**

In figure 6 left shows a Kp index greater 5 on the 286th day. The daily VLF amplitude on the top right for NAA-NTZ varies little during the disturbed time, but the overall level does not change significant. We conclude, that these disturbance is not a concern for the determination of the October effect. For NAA-SOD in figure 6 middle the VLF daily amplitude on the right does not show a clear disturbance on that day. Adding to that, disturbance occurs in the middle of October effect. Therefore, the determination of the ending and starting of the October effect should not be affected. In figure 6 bottom right the VLF signal amplitude for NRK-STJ, shows a small distinct decrease just before 15:00 UT on the disturbed day. But here it can also be seen that on the day prior the transmitter had an outage, which leads to a much greater change in the signal amplitude. As these things may occur, we decided to use such a long smoothing windows of 21 days with the rolling mean, to ensure, that these things do not influence the determination of the October effect. These outages have potentially more impact on the method and are therefore removed before smoothing.

[Figure]

Figure 6: Left: The course of the Kp index during the duration of the observed October effect in that propagation path. Right: The daily amplitude one day prior (blue), at the day of the enhanced Kp index (black) and one day after (orange).

Concluding this analysis, only 8 from the 14 potential disturbed links show a significant change in VLF signal amplitude in the disturbed time. However, these disturbances are only visible in one day each and for the more disturbed days happen in the middle of the October effect. The propagation paths, which have the disturbance in the middle of the October effect, also have values for several years, which makes them additional more robust. As the paper focuses more on the beginning and ending time of the October effect, an significant influence on the determination of the timing of the October effect is unlikely. From these disturbed links only one shows to have had an affect on the determination of the starting time of the October effect for the propagation path of NDK-KIL in 2016. The overall trends are not affected by this one data point out of 63 total (counting also the undisturbed). As we show the spread of $t_{start}$ and $t_{end}$ for each propagation path over the years, we do not want to exclude disturbances per se, but rather we want to show natural variations, i.e. due to atmospheric or solar influences.

**Comment 2**

Discussion is very superficial. I understand that there is no consensus about the likely mechanism to explain the October effect. Physically, we believe that it could be the result of the modification in the atmosphere since the ionospheric condition remains stable during those periods. The paper by Macotela et al. (2021 and references therein) explores those possibilities that can be used by the authors to improve the discussion.

**Answer 2**

Our idea for this paper was to statistically investigate the October effect and its occurrences. Together with other researchers at our institute, we try to find the physical explanation of the October effect, which than is planned to be presented in an additional paper. For example, in the recently published paper [Wendt, V., Schneider, H., Banyś, D., Hansen, M., Clilverd, M. A., & Raita, T. (2024). Why does the October effect not occur at night? Geophysical Research Letters, 51, e2023GL107445. https://doi.org/10.1029/2023GL107445], it is discussed why the October effect does not occur in the nighttime. As the October effect is a broad topic, which includes investigating possible correlation to neutral dynamics in the atmosphere and combining all this to model the October effect with the Long-Wavelength Propagation Capability (LWPC), this paper aims to create a basis for this ongoing investigation.

**Report 3**

**Comment 1**

The paper describes the "October effect" as a sharp decrease in VLF signal amplitude but lacks a clear physical explanation for this phenomenon. This absence of a more in-depth explanation, supported by previous studies, limits understanding the physics underlying this effect. Further exploration of the mechanisms contributing to the October effect is needed.

**Answer 1**

Our idea for this paper was to statistically investigate the October effect and its occurrences. Together with other researchers at our institute, we try to find the physical explanation of the October effect, which than is planned to be presented in an additional paper. For example, in the recently published paper [Wendt, V., Schneider, H., Banyś, D., Hansen, M., Clilverd, M. A., & Raita, T. (2024). Why does the October effect not occur at night? Geophysical Research Letters, 51, e2023GL107445. https://doi.org/10.1029/2023GL107445], it is discussed why the October effect does not occur in the nighttime. As the October effect is a broad topic, which includes investigating possible correlation to neutral dynamics in the atmosphere and combining all this to model the October effect with the Long-Wavelength Propagation Capability (LWPC) Code, this paper aims to create a basis for this ongoing investigation.

**Comment 2**

Using derivatives to identify the "October effect" may be highly sensitive to noise and short-term data variations, increasing the risk of false positives or negatives in detection, especially given the use of a predefined threshold. Does the methodology incorporate any form of supervised classification, or how do the authors address this potential sensitivity to noise?

**Answer 2**

To mitigate any influence of noise to the determination of the October effect, we use multiple instances of smoothing and the noise robust median. Additionally, the disturbances are mainly short term, which may only affect the VLF signal for a short time. The idea to use derivatives to determine the October effect, makes the method insensitive about the general signal level. To even mitigate the effect of noise even more, in the future we will rely on applying our method on the dataset from Schneider, H., Wendt, V., Banys, D., Clilverd, M., & Raita, T. (2024). Processing of VLF amplitude measurements: Deduction of a quiet time seasonal variation. Radio Science, 59, e2023RS007834. https://doi.org/10.1029/2023RS007834. There, disturbances are filtered out, but as this "level 2" dataset was being developed while this analysis was conducted, we did not use it in this study. However, we also compared both outputs for the determination of the October effect, and we draw the same conclusions.

**Comment 3**

The authors adopt a series of criteria to identify the October effect, including a threshold of 0.005 dB/day$^2$. Given the challenges mentioned in the text, this raises questions about the reproducibility of the methodology. Since VLF measurements can be influenced by various external factors (e.g., solar flares, geomagnetic storms) that may cause amplitude fluctuations, it is unclear how this specific threshold was determined. Additionally, would this threshold remain valid for studying the October effect during periods of intense geomagnetic storms, or should data from such periods be excluded? Clarification on these points would strengthen the reproducibility of the detection method.

**Answer 3**

Thank you for this hint, as geomagnetic storm could potentially have an affect on the VLF signal. Only a handful of moderate geomagnetic storms occurred during the investigation period and moreover these did not significantly affect the results. Therefore, we did not exclude such periods. As we show the spread of $t_{start}$ and $t_{end}$ for each propagation path over the years, we do not want to exclude disturbances per se, but rather we want to show natural variations, i.e. due to atmospheric or solar influences. For an extensive investigation on how geomagnetic storm influence the determination of the October effect in our data set, please refer to Answer 1 in Report 2, for more information. Solar flares affect the VLF signal only for a short duration, therefore such disturbances are mitigated by the long smoothing period. For example in 3 bottom right, a solar flare is visible at around 14:00 for NDK-SOD 2015. The effect only last about 1 hour, which would only be half the observation time at local noon for out method (In this example the solar flare is not even in the observed noon time window). As this long smoothing period and the additional smoothing done in the derivatives, such disturbances do not exceed the threshold. The threshold of 0.005 dB/day$^{-2}$ is an empirical value derived from handling the dataset. Since, we apply strong smoothing and work with derivatives the relative range in the signal amplitude is comparable for different propagation paths, so the threshold can be set empirically.

**Minor Comments:**

1. Line 86: Change "doesn't" to "does not" for a more formal tone.

2. Figure 1 caption: Correct "propagations paths" to "propagation paths."

3. Line 97: Correct the typo "evets" to "events."

4. Line 135: Change "show" to "shown".

We thank the reviewer for spotting these mistakes and the hints to improve to language. We have addresed all the proposed changes in the text.